Chromatin accessibility complex subunit 1 enhances tumor growth by regulating the oncogenic transcription of YAP in breast and cervical cancer

Li Shasha 1
Wang Lulu 2
Shi Jing 3
Chen Yi 1
Xiao Ang 1
Huo Bingyue 1
Tian Wenjing 1
Zhang Shilu 1
Yang Gang 3
Gong Wensheng 3
Zhang Huixia hxzhang@hust.edu.cn 1
1 Department of Human Anatomy, School of Basic Medicine, Tongji Medical College, Huazhong University of Science and Technology , Wuhan , China
2 Department of Pediatrics, Union Hospital, Tongji Medical College, Huazhong University of Science and Technology , Wuhan , China
3 Xiangyang Center for Disease Control and Prevention , Xiangyang , China
Tyagi Abhishek
Electronic publication date: 2024 Jan 10
Publication date: 2024
Volume: 12
Electronic Location ID: e16752
Received 2023 Aug 23; Accepted 2023 Dec 13
Copyright: ©2024 Li et al.
Copyright year: 2024
Copyright holder: Li et al.
License: This is an open access article distributed under the terms of the Creative Commons Attribution License, which permits unrestricted use, distribution, reproduction and adaptation in any medium and for any purpose provided that it is properly attributed. For attribution, the original author(s), title, publication source (PeerJ) and either DOI or URL of the article must be cited.
License URL: https://creativecommons.org/licenses/by/4.0/

Keywords: CHRAC1, Breast cancer, Cervical cancer, Hippo pathway, YAP, Oncogenictranscription

Funding: National Natural Science Foundation of China #82273064 #81902853 Natural Science Foundation of Hubei Province 2022CFB202 This study was supported by the National Natural Science Foundation of China (grant numbers: #82273064, #81902853) and the Natural Science Foundation of Hubei Province (grant numbers: 2022CFB202). The funders had no role in study design, data collection and analysis, decision to publish, or preparation of the manuscript.

==============================
Background

As a component of chromatin remodeling complex, chromatin accessibility complex subunit 1 (CHRAC1) is critical in transcription and DNA replication. However, the significance of CHRAC1 in cancer progression has not been investigated extensively. This research aimed to determine the function of CHRAC1 in breast and cervical cancer and elucidate the molecular mechanism.

Methods

The Bio-ID method was used to identify the interactome of transcriptional activator Yes-associated protein (YAP) and the binding between YAP and CHRAC1 was verified by immunofluorescence. CCK8, colony formation and subcutaneous xenograft assays were conducted to explore the function of CHRAC1 in cancer cell proliferation. RNA-seq analysis and RT-PCR were used to analyze the transcription program change after CHRAC1 ablation. The diagnostic value of CHRAC1 was analyzed by TCGA database and further validated by immunohistochemistry staining.

Results

In the current study, we found that the chromatin remodeler CHRAC1 was a potential YAP interactor. CHRAC1 depletion suppressed breast and cervical cancer cell proliferation and tumor growth. The potential mechanism may be that CHRAC1 interacts with YAP to facilitate oncogenic transcription of YAP target genes in Hippo pathway, thereby promoting tumorigenesis. CHRAC1 was elevated in cervical and breast cancer biopsies and the upregulation correlated with shorter survival, poor pathological stages and metastasis of cancer patients. Moreover, CHRAC1 expression was statistically associated with YAP in breast and cervical cancer biopsies.

Conclusions

These findings highlight that CHRAC1 contributes to cancer progression through regulating the oncogenic transcription of YAP, which makes it a potential therapeutic target for cancer treatment.

Introduction

According to the latest WHO Cancer Survey report for 2020, there are about 10 million cancer deaths and 19 million new cases worldwide (Sung et al., 2021). At present, radiotherapy and chemotherapy have a certain killing effect on cancer cells, but the side effects are relatively obvious (De Ruysscher et al., 2019; Johnston & Beckman, 2019; Petit et al., 2021; Wang et al., 2021b). With the spread of genetic testing and the development of targeted drugs, targeted therapies can be applied to cancer patients whose driver genes are mutated, amplified or rearranged (Wu et al., 2021; Yang et al., 2020). This has significantly extended the survival time of cancer patients. However, over time, resistance to targeted drugs greatly limits their efficacy and application (Aldea et al., 2021). Therefore, it is very important to deeply explore the pathological process of cancer and develop new therapeutic targets.

YAP is a crucial downstream transcriptional coactivator in Hippo signaling pathway (Driskill & Pan, 2021). YAP can bind to TEAD family protein to activate transcription of downstream targets which are implicated in cell growth and differentiation (Ardestani, Lupse & Maedler, 2018; Lim et al., 2014; Pan et al., 2018). Data from previous reports suggest the vital function of YAP/TAZ in transcription regulation of cancer cells (Chang et al., 2018; Cordenonsi & Piccolo, 2018; Galli et al., 2015; Oh et al., 2014b; Skibinski et al., 2014; Stein et al., 2015; Zanconato et al., 2015). Meanwhile, studies have also shown that the disturbance of YAP signal probably is the main mechanism of drug resistance to diversified targeting and chemotherapy, and the expression of PD-L1 and various cytokines mediated by YAP is very important for tumor immune escape (Nguyen & Yi, 2019). These studies imply that YAP may be used as a potential target for cancer and supply an effective idea for clinical treatment (Cunningham & Hansen, 2022).

The ATP-dependent chromatin remodeling complexes play a crucial role in altering the structure of chromatin and controlling the accessibility of transcription factors to DNA, thereby facilitating the dynamic regulation of gene expression in the context of cancer progression. As a chromatin remodeling factor, chromatin accessibility complex subunit 1 (CHRAC1) has been involved in transcription and DNA replication (Clapier et al., 2017; Hasan & Ahuja, 2019). According to the result of a small interfering RNA screening, CHRAC1 was identified as an oncogenic driver gene (Mahmood et al., 2014). In addition, CHRAC1 expression level was increased in cisplatin resistant ovarian cancer cell lines and lung cancer tumor tissues, and CHRAC1 disruption inhibited the migration and invasion of cisplatin resistant ovarian cancer cells and H1299 lung cancer cells (Wang et al., 2021a; Yang et al., 2021). Although previous studies have reported the significance of CHRAC1 in the progression of several cancers, the mechanism has not been well studied and the role of CHRAC1 in other cancers remains to be elucidated.

Although it has been well established that YAP activates transcription by recruiting the transcription factor TEAD, whether chromatin remodeling complex is involved in this process remained unclear. In the current research, we identified CHRAC1 as a potential YAP interactor using the Bio-ID method and demonstrated that CHRAC1 regulated cell proliferation through promoting the oncogenic transcription of YAP target oncogenes in breast and cervical cancer. Together, our findings reveal that CHRAC1 plays a crucial role in enhancing YAP transcription activity and tumorigenesis, suggesting it may be a potential target for cancer therapy.

Material and Methods

Cell line and transduction

HEK293T, MDA-MB-231 and Hela cells were purchased from Procell (Wuhan, China) and cultured in DMEM (Gibco) with 10% FBS (Gibco). For CHRAC1 knockdown, cancer cell was transduced with pLKO.1-puro (shNC) or pLKO.1-puro-shCHRAC1 (1# or 2#). Interfering sequences were: shCHRAC1-1#: 5′-ACTCCACTGTCTCTAAGTAAA-3′; shCHRAC1-2#: 5′-ACTCCACTGTCTCTAAGTAAA-3′.

Western blotting and Co-immunoprecipitation (Co-IP) assay

Cells were lysed in Beyotine lysis buffer (Cat#: P1003) containing protease inhibitor and phenylmethylsulfonyl fluoride (PMSF) at 4 °C for 20 min and centrifuged at 4 °C, 12000 rpm for 15 min. Then cell lysate was denatured by boiling with SDS sample buffer at 95 °C for 10 min and separated by 10% SDS-PAGE gel. After that, the membranes were incubated with specific primary antibody (ABclonal rabbit CHRAC1: A14896, 1:500; ABclonal rabbit CHRAC17: A6469, 1:1000) overnight at 4 °C, secondary antibody for 1 h at room temperature and analyzed with electrogenerated chemiluminescence (ECL) system (Cat#: WBKLS0500; Millipore, Burlington, MA, USA).

For Co-IP assays, cell lysates of 293T cells transfected with Flag-tagged CHARC1and HA-tagged YAP vectors were incubated with Agarose beads (Cat#: 16-266; Millipore, Burlington, MA, USA) and Flag antibody (E005; ABclonal) or IgG control overnight at 4 °C. The agarose beads/protein complex was washed 4 times with NP40 buffer and denatured at 95 °C for 15 min with SDS sample buffer and then subjected SDS-PAGE.

Bio-ID

Bio-ID experiment was conducted with reference to previous method (Zhang et al., 2018a). Briefly, cells that stably expressed YAP-BirA* were cultured in complete medium containing 50 µM of biotin for 24 h, then washed with pre-cooled 1 x PBS 6-8 times, lysed with 1ml lysis buffer including protease inhibitor, and sonicated. The supernatant was obtained by centrifugation (4 °C, 15,000 rpm, 10 min) and incubated with 50 µl Streptavidin-conjugated beads (17511301; GE Healthcare, Chicago, IL, USA) overnight at 4 °C. After washing with 2% sodium dodecyl sulfate (SDS), 1% Triton X-100, 250 mM LiCl, 50 mM Tris buffer and 50 mM NH4HCO3 buffer, the beads were stored at −20 °C for mass spectrometry or western blotting analysis. For western blotting detection, beads were denatured in 2X loading buffer at 100 °C for 10 min. Then supernatants were subjected SDS-PAGE. After blocking with 3% Bovine albumin (BSA) for 2 h at room temperature, the membranes were incubated with HRP-conjugated Streptavidin for 1 h at room temperature. After washing 4 times with TBST, the membranes were analyzed with electrogenerated chemiluminescence (ECL) system.

Survival analysis of CHRAC1

The Gene Expression Profiling Interactive Analysis version 2 (GEPIA2) (http://gepia2.cancer-pku.cn/) was used to investigate the overall survival (OS) and dis-ease-free survival (DFS) curves, as well as a survival map of CHRAC1 in The Cancer Genome Atlas (TCGA) tumor types. The expression threshold was set at 50% for high CHRAC1 expression and low CHRAC1 expression.

CCK-8, wound-healing and colony formation assays

Cell proliferation curves were measured by cell Counting Kit-8 (CCK-8) (BMU106-CN; Abbkine, Wuhan, China). Cells were seeded in 96-well plate with 4,000 cells in each well and cultured for 0, 24, 48, 72 and 96 h. Then the 450 nm absorbance was measured at each time point by adding CCK-8 reagent. For wound-healing assay, cells were inoculated in 6-well plates with 70% confluence and cultured for 36 h. Cell monolayer was scratched with the pipette tip to generate wound which was photographed every 12 h to monitor cell migration. For colony formation, cells were inoculated in 6-well plate with 1,000 (Hela) or 4,000 (MDA-MB-231) cells each well. After 10 days, the clones were stained with 0.05% crystal violet for imaging to monitor cell growth.

Real-time quantitative PCR (RT-qPCR)

RT-qPCR was used to verify the effect of CHRAC1 knockdown on YAP target genes in MDA-MB-231 and Hela cells. In this study, shNC was used as a control group and CHRAC1 shRNA (1# and 2#) was used as experimental group. Total RNA from cells was extracted according to the instructions of Vazyme RNA extraction Kit (Cat#: RC112-01; Vazyme, Beijing, China). RNA concentration and Purity were detected by Nanodrop spectrophotometer and 1µg RNA was applied to reverse transcription to synthesize complementary DNA (cDNA) using Reverse Transcription Kit (Cat#: RK20429, ABclonal) under the following conditions: 37 °C for 2 min, 55 °C for 15 min, 85 °C for 5 min. The reverse-transcribed cDNA was diluted 5-fold and RT-qPCR was conducted with the SYBR Green qPCR Mix reagent (Cat#: RK21203, ABclonal) on a Bio-Rad quantitative PCR instrument under the following conditions: 95 °C for 3 min, followed by 40 cycles of 95 °C for 5 s, 60 °C for 30 s. Glyceraldehyde-3-phosphate dehydrogenase (GAPDH) was used as reference gene. The method to calculate the relative expression was 2−ΔΔCt. All of the RT-qPCR reactions were performed in duplicates. RT-qPCR primers sequences are available in supplementary materials.

Immunofluorescence (IF)

Cells were cultured for 36 h, fixed with 4% paraformaldehyde, and blocked with Image-iT signal enhancer (Thermo: I36933). Thereafter, cells were incubated with primary antibody (Santa Cruz mouse YAP: sc-101199, 1:200; ABclonal rabbit CHRAC1: A14896, 1:200) overnight at 4 °C and secondary antibody for 1 h. For Ki67 staining, freshly dissected tumor tissue was fixed and embedded in OCT reagent (Sakura) and cut to 4µm sections. Thereafter, sections were treated with citrate buffer for antigen retrieval and the following steps are similar to cell immunofluorescence staining.

RNA-seq

After the extraction of total RNA, the sequencing library was generated using the Prep Kit V2 (Illumina, San Diego, CA, USA) for RNA Library according to the manufacturer’s procedure. RNA-seq library was sequenced using the 75-nucleotide paired-end sequencing protocol on a NextSeq sequencer (Illumina). The technical replicates are duplicates for each sample and the RNAseq details are as follows: sequencing depth: 34.10939x; dispersions: 0.017797; effect 2; false positive rate: 0.05. The statistical power of this experimental design, calculated in RNASeqPower is 0.98. For bioinformatics analysis: genes whose Fold Change (padj <= 0.05) is less than or equal to -2 were defined as down-regulated genes (CHRAC1 knockdown versus control cell). Heat-map for different cancer hallmarks was created by TB-tools. The sequencing data have been deposited in the FigShare (DOI: 10.6084/m9.figshare.23989101).

Mouse model

Five-week-old male and female BALB/c nude mice (60 single), there was a lack of mature T lymphocytes, were purchased from Animal Center in Tongji Medical College, and subsequently housed under specific-pathogen-free conditions (Temperature: 22  ± 2 °C, Relative humidity: 60  ± 10%, 12 h light/dark cycle) at the Animal Center in Tongji Medical College. All the mice were given an unrestricted access of sterile food and water. Nine single male/female mice were randomly assigned to three separate groups (each group: three singles (sample size is determined according to the results of the preliminary experiment)): shNC group (control group) and CHRAC1 shRNA group (1# and 2#) (experimental group). 5 × 106 MDA-MB-231 or Hela cells treated with shNC or CHRAC1 shRNA were injected into the dorsal side of each nude mice (MDA-MB-231 cell: female mice; Hela cell: male mice). After about 2 weeks, tumor volume was monitored through double blind trials with vernier caliper every other day, and the size was measured as V = (L × W2) / 2 (L: length, W: width). Tumors were removed, imaged and weighed from mice with no significant weight change after 35 or 40 days. The tumor volume is generally no larger than 2000 mm3. The use and research design of animals was approved the Institutional Review Committee of Huazhong University of Science and Technology. The approval number: 2022 IACUC Number: 3148.

Immunohistochemistry (IHC) staining

Tissue sections of 48 human breast cancer and 48 cervical cancer specimens were purchased from Shanghai Wellbio Biotechnology company (Shanghai, China). Verbal informed consent was obtained and cancer specimens in this research have been approved by the Ethics Committee of of Huazhong University of Science and Technology and Shanghai Zhuoli Biotechnology Co., LTD (approval number: ZL2019-9-LL028, ZL2019-11-LL029). IHC staining was applied to detect the expression levels of CHRAC1 and YAP in breast and cervical cancer tissues. After deparaffinization and rehydration, antigen retrieval was performed. Endogenous peroxidase was blocked with 10% goat serum 1 h. Next, tissue microarrays were stained with primary antibodies (rabbit CHRAC1: Abclonal A14896, 1:200; rabbit YAP: Cell Signaling Technology 14074, 1:200) overnight at 4 °C and secondary antibody for 1 h. After washing, color development was performed with DAB kit (Maxim, DAB-0031) and the microarray was scanned with 3DHISTECH’s Slide Converter. The scanned images were opened with the Caseviewer software and enlarged to different magnifications. Click the “take snapshot” button on the software to take a picture. The images were saved in PNG format with 300 dpi resolution.

Statistical analysis

Statistical analysis was conducted with Prism 8 (GraphPad Software, La Jolla, CA, USA). Two-way analysis of variance (ANOVA) was used in CCK-8 and tumor volume detection assay. Student’s t-test (two-tailed) was performed to compare CHRAC1 expression, wound-healing ability, colony formation ability, tumor weight and Ki67 staining. Linear regression analysis was performed to investigate the correlation between CHRAC1 and YAP in cancer tissues.

Results

Interactome of YAP and CHRAC1 is a potential Bio-ID interactor of YAP

To clarify YAP oncogenic regulators, we identified the spatial proximity interactome of YAP using the Bio-ID (proximity-dependent biotin identification) method (Fig. 1A). HEK293T cells expressing YAP-BirA*-HA were cultured in medium supplemented with or without biotin; much more biotinylated proteins were captured in total cell lysates cultured with biotin (Figs. 1B and 1C). Following mass spectrometry, we identified 254 potential interactors of YAP (YAP Bio-ID interactor). The Gene Ontology (GO) analysis on biological process for YAP Bio-ID interactors revealed a significant enrichment in Hippo pathway, cell division, cell–cell adhesion, DNA repair, DNA replication, mRNA processing and chromatin remodeling (Fig. 1D). In additional to numerous known YAP interactors (AMOT, LATS1 and AMOTL1), CHRAC1 attracted our attention. As a member of chromatin remodeling complex, CHRAC1 has been involved in transcription and DNA replication. However, the significance of CHRAC1 in cancer development has not been investigated extensively. Therefore, we focused on the study of CHRAC1.

Figure 1 Interactome of YAP and CHRAC1 is a potential Bio-ID interactor of YAP.

(A) Flowchart of Bio-ID approach. (B) Schematic illustration of YAP-BirA*-HA construct. (C) HEK293T cells stably expressing YAP-BirA*-HA construct were cultured with 50 µM/L biotin for 24 h. The biotinylated proteins were captured by streptavidin-agarose beads and detected by HRP-conjugated streptavidin. (D) GO analysis on biological process for YAP Bio-ID interactors.

High expression of CHRAC1 predicts advanced pathological tumor stages and poor survival

To explore the expression of CHRAC1 in cancer tissues, we analyzed the expression of CHRAC1 in cancer tissues including breast and cervical. Immunohistochemistry (IHC) result displayed that CHRAC1 was mainly expressed in the nucleus and the staining was extremely weak in para-tumor tissues, while breast and cervical cancer biopsies showed strong CHRAC1 signal (Figs. 2A and 2B).

Figure 2 High expression of CHRAC1 predicts advanced pathological tumor stages and poor survival in breast and cervical cancer.

(A) Representative CHRAC1 IHC staining in breast and cervical cancer tumor tissues and para-tumor tissues. Scale bar, 20 µm and 5 µm. (B) Statistical analysis of CHRAC1 intensity was shown in diagram. (C–E) Overall survival curves of patients with high or low expression of CHRAC1 in CESC (C), KIRP (D) and LAML (E). Data were from TCGA database and were analyzed with GEPIA2 tool. (F) Representative IHC staining of CHRAC1 in different tumor stages of breast cancer biopsies. Scale bar, 50 µm and 5 µm. (G) Quantification of CHRAC1 staining in Fig. 2F was demonstrated in diagram. (H) Quantification of CHRAC1 staining in cervical cancer biopsies with metastasis was demonstrated in the histogram. * P < 0.05, *** P < 0.001. Two-tailed student’s t-test.

To evaluate the potential clinical value of CHRAC1, we explored the correlation between CHRAC1 expression and patient survival within different tumors in the TCGA dataset (Fig. S1A). High CHRAC1 mRNA level was correlated with decreased overall and disease-free survival in cases of CESC and other tumor types by using the Gene Expression Profiling Interactive Analysis version 2 (GEPIA2) tool (Figs. 2C–2E, Figs. S1B–S1K). Additionally, the CHRAC1 level was significantly associated with pathological stages of kidney renal clear cell carcinoma (KIRC), adrenocortical carcinoma (ACC) and PAAD patients (Fig. S2). Moreover, in breast cancer specimens (n = 48), CHRAC1 expression increased with the progression of tumor stage (Figs. 2F and 2G). In cervical cancer specimens (n = 48), patients with metastasis had elevated CHRAC1 protein expression compared to those patients with primary tumors (Fig. 2H). Together, these data manifest that high CHRAC1 expression may predict advanced pathological tumor stages and poor survival in breast and cervical cancer.

Downregulation of CHRAC1 inhibits tumor growth

To validate the effect of CHRAC1 on cancer progression, we silenced CHRAC1 (Fig. 3A) in breast and cervical cancer cell lines (MDA-MB-231 and Hela). The downregulation of CHRAC1 have been confirmed by immunoblot, the result showed that the knock down of CHRAC1 specifically inhibited the protein level of CHRAC1 but not another family member CHRAC17 (Fig. S3A). CCK-8 assay displayed that CHRAC1 silencing suppressed the activity of MDA-MB-231 and Hela cells (Fig. 3B). Wound-healing assay also indicated that CHRAC1 down-regulation restrained cancer cell migration (Fig. 3C). In addition, colony formation assay showed that cancer cells with ablated CHRAC1 demonstrated significantly decreased colony size and number (Fig. 3D). To make the knockdown data more compelling, we also investigated the effect of CHRAC1 over-expression on cancer cells. The results showed that CHRAC1 over-expression significantly promoted cell migration and proliferation both in MDA-MB-231 and Hela cells (Figs. S3B–S3D). Collectively, these data suggest that CHRAC1 silencing inhibits the proliferation of breast and cervical cancer cells.

Figure 3 Downregulation of CHRAC1 inhibits tumor growth.

(A) Silencing of CHRAC1 in MDA-MB-231 and Hela cells detected by RT-qPCR. (B) CHRAC1 silencing restrains cell viability. (C) CHRAC1 silencing restrains cell migration. The statistical analysis was demonstrated in histogram. (D) The effect of CHRAC1 silencing on colony formation. The statistical analysis of clone numbers was demonstrated in histogram. *** P < 0.001, ** P < 0.01, * P < 0.05. (E) Typical tumor images in mice inoculated with control and CHRAC1-silenced cancer cells. (F) CHRAC1 silencing reduces tumor volume. **** P < 0.0001, two-way ANOVA. (G) CHRAC1 silencing reduces tumor weight. ** P < 0.01, Student’s t-test (two-tailed). (H) Ki67 staining for proliferating cells in control and CHRAC1 silenced Hela xenografts. Quantification of Ki67-positive cells was demonstrated in the histogram. *** P < 0.001, ** P < 0.01, Student’s t-test (two-tailed).

To verify the influence of CHRAC1 on tumorigenesis, we conducted a xenograft assay in nude mouse. Downregulation of CHRAC1 in Hela cells could significantly reduce tumor size, and tumor formation was not possible in MDA-MB-231 cells with CHRAC1 inhibition (Figs. 3E and 3F). The average weight of CHRAC1-silenced Hela xenografts was about one-third of the control group (Fig. 3G). To investigate the proliferation capacity of tumor cells, we performed immunofluorescence staining of Ki67 for tissue sections. The ratio of Ki67 positive cells was statistically reduced in CHRAC1-silenced tumors compared to control tumors (Fig. 3H). Collectively, these data display that CHRAC1 silencing suppresses tumor growth of breast and cervical cancer cells in nude mice.

Inhibition of CHRAC1 suppresses the oncogenic transcription of YAP

To illustrate the mechanism of CHRAC1 regulation on tumorigenesis, we performed RNA-seq assay in CHRAC1-silenced Hela cells. CHRAC1 ablation caused differential expression of 2,595 genes, of which 1,300 genes were increased and 1,295 genes were decreased (Fig. 4A). Gene ontology (GO) annotation of the 1,295 down-regulated genes indicated that they were highly enriched for GO terms linked to transcription, cell proliferation, cell apoptosis, cell migration and related signaling pathways (Fig. 4B). A heatmap of RNA-seq data clearly showed that knock-down of CHRAC1 suppressed the expression of representative cancer hallmarks, such as DNA repair, G2M checkpoint and the P53 pathway (Fig. 4C). Additionally, Kyoto Encyclopedia of Genes and Genomes (KEGG) annotation displayed that CHRAC1 silencing resulted in significant downregulation (p < 0.01) of Hippo pathway-related genes (Fig. 4D). Then, we focused on the direct YAP target oncogenes previously identified (Zanconato et al., 2018; Zanconato et al., 2015). Gene set enrichment analysis (GSEA) displayed that YAP target gene signature was enriched in control Hela cells but not in CHRAC1 silenced cells, suggesting that CHRAC1 might trigger the oncogenic transcription program of YAP in Hela cells (Fig. 4E). Further, RT-qPCR results confirmed that depletion of CHRAC1 reduced the mRNA level of three classical YAP target genes (CTGF, CYR61, ANKRD1) (Fig. 4F). We also performed immunofluorescence (IF) assay for CTGF in Hela and MDA-MB-231 cell lines to validate our finding at the protein level. The results showed that knock down of CHRAC1 significantly decreased the fluorescence intensity of CTGF both in Hela and MDA-MB-231 cells (Fig. 4G). Together, these results demonstrate that CHRAC1 indeed influences the transcriptional activation of YAP.

Figure 4 Inhibition of CHRAC1 suppresses the oncogenic transcription of YAP.

(A) The volcano map displayed the differential expression of genes upon CHRAC1 inhibition. (B–D) GO annotation (B), heatmap (C), and KEGG analysis (D) of the down-regulated genes upon CHRAC1 silencing. (E) GSEA analysis of YAP targets in CHRAC1 knockdown Hela cells. (F) Effects of CHRAC1 knockdown on mRNA expression of YAP targets in MDA-MB-231 and Hela cells. (G) Effects of CHRAC1 knockdown on protein level of YAP target CTGF in MDA-MB-231 and Hela cells by IF staining.

CHRAC1 interacts with YAP and is positively correlated with YAP in breast and cervical cancer patients

According to the Bio-ID results, CHRAC1 may be a potential interactor of YAP. To validate the interaction of CHRAC1 and YAP, we performed exogenous co-immunoprecipitation (Co-IP) assay. The result showed that flag-tagged CHARC1 was able to specifically precipitated HA-tagged YAP in 293T cells compared to the IgG control (Fig. 5A, left panel). In addition, Co-IP detection was also conducted in CHRAC1 overexpressed MDA-MB-231 and Hela cells, and it was found that YAP could specifically co-purify CHRAC1 in both cells (Fig. 5A, right panel). Furthermore, immunofluorescence assay displayed that CHRAC1 co-localized with YAP both in MDA-MB-231 and Hela cells (Fig. 5B). Therefore, there is indeed an interaction between YAP and CHRAC1. Then, we investigated the clinical correlation of CHRAC1 and YAP and found CHRAC1 was associated with YAP across Pan-cancer, including BRCA and CESC in TCGA database (Figs. 5C–5E). In addition, both breast and cervical cancer biopsies with high CHRAC1 expression showed stronger YAP staining (Fig. 5F). Consistent with this, there was significant correlation ( p < 0.0001) between YAP and CHRAC1 in these cancer biopsies (Fig. 5F). Thus, CHRAC1 is elevated in breast and cervical cancer and the upregulation correlates well with YAP.

Figure 5 CHRAC1 interacts with YAP and is positively correlated with YAP in breast and cervical cancer patients.

(A) Exogenous Co-IP of HA-YAP and Flag-CHRAC1 in HEK293T cells (left panel). Co-IP detection of YAP and CHRAC1 in CHRAC1 overexpressed MDA-MB-231 and Hela cells (right panel). (B) CHRAC1 is co-localized with YAP in MDA-MB-231 (upper panel) and Hela (lower panel) cells. (C–E) Correlation analysis between YAP and CHRAC1 in pan-cancer (C), BRCA (D) and CESC (E) according to the TCGA database. (F) Typical IHC images of breast and cervical cancer specimens with low and high YAP/CHRAC1 expression. Scale bar, 50 µm and 5 µm. Correlation analysis of YAP and CHRAC1 in breast and cervical cancer tissues was shown in diagram.

In summary, our study found that depletion of CHRAC1 suppresses cancer cell proliferation and tumor growth. The potential mechanism may be that CHRAC1 interacts with YAP to enhance the transcription of YAP down-stream oncogenes to promote tumor growth (Fig. 6). Moreover, CHRAC1 is frequently upregulated in diverse cancers and the upregulation is statistically associated with YAP activation and poor prognosis in cancer patients. These findings highlight CHRAC1 might be a promising candidate for cancer diagnosis and therapy.

Figure 6 Mechanism of CHRAC1 promoting cancer cell proliferation.

The potential mechanism may be that CHRAC1 interacts with YAP to enhance the transcriptional activation of YAP downstream target genes, such as CTGF, CYR61 and ANKRD1, and thus promotes the tumor growth.

Discussion

CHRAC1, a component of the chromatin remodeling complex, is associated with poor prognosis of cancer patients (Poot et al., 2000). In addition, researches have reported that abnormal expression of CHRAC1 is frequently associated with the occurrence and progression of human cancer (Wang et al., 2021a; Yang et al., 2021). However, the mechanism remains to be clarified and the function of CHRAC1 in various cancer types has not been fully studied.

In this study, we identified CHRAC1 as a potential oncogene in multiple human tumors via pan-cancer analyses and experimental verification. First of all, we discovered that CHRAC1 was upregulated in a variety of cancer tissues compared to relevant normal tissues. Meanwhile, CHRAC1 expression was statistically correlated with poor survival and pathological stages in various tumor types. Collectively, these findings indicate that elevated CHRAC1 may be a prognostic factor for cancer.

Next, we verified the oncogenic function of CHRAC1 through experiments. In in vitro study, we discovered that knockdown of CHRAC1 restrained the growth of breast and cervical cancer cells. However, there are some limitations to this research. For example, it may make the knock down data compelling to investigate the effects of CHRAC1 overexpression on cell growth. In in vivo study, we demonstrated that CHRAC1 silencing decreased tumor weight and tumor size in MDA-MB-231 and Hela xenograft mouse model. However, whether CHRAC1 affects tumor metastasis needs further study. Additionally, development of drugs that target CHRAC1 may advance the findings to clinical applications.

To clarify how CHRAC1 affects cancer development, we conducted RNA-seq assay and found that the transcription program of control cells and CHRAC1 knockdown cells was significantly different. The inhibition of CHRAC1 also affected the expression level of many cancer hallmarks. Moreover, many known YAP target genes were significantly downregulated in CHRAC1 knockdown groups. It has been well established that YAP target genes served as oncogenes in many cancer types (Zanconato, Cordenonsi & Piccolo, 2016; Zhang et al., 2018b). These indicate that CHRAC1 may promote tumor growth by enhancing the oncogenic transcription of YAP. However, in addition to YAP, whether CHRAC1 knockdown affects other transcription factors remains to be studied.

Base on the Bio-ID results, CHRAC1 was a potential interactor of YAP and the interaction between these two proteins was verified by Co-IP and IF assays. In addition, numerous known YAP interactors (AMOT, LATS1 and AMOTL1) have been identified by Bio-ID method, which confirms the feasibility of this approach. According to previous reports, YAP is known to recruit transcription factors TEAD1-4, RNA polymerase, the mediator complex, and other factors to establish a transcription hub within the nucleus, facilitating oncogenic transcription during cancer progression (Oh et al., 2014a; Stein et al., 2015; Zanconato et al., 2015). Notably, there are robust interactions between these components and YAP. Our findings demonstrate the presence of CHRAC1 in YAP complex, exhibiting strong co-localization with YAP within the nucleus. Moreover, the loss of CHRAC1 impedes the transcriptional activation of YAP. Consequently, we suggest that the interaction between YAP and CHRAC1 plays a crucial role in recruiting CHRAC1 to YAP transcription hub, thereby promoting the oncogenic transcription.

To further investigate the clinical significance of CHRAC1, we conducted IHC analysis in human breast and cervical cancer biopsies. The expression of CHRAC1 in cancer tissues was significantly higher than that in the para-cancer group. Moreover, YAP was also highly expressed in breast and cervical cancer specimens with high expression of CHRAC1. Consistent with this, there was a statistical correlation between YAP and CHRAC1 (p < 0.0001). Despite this, specimens from different cancer types are required to verify the association between CHRAC1 and YAP. Therefore, the practical application of CHRAC1 in cancer prediction and treatment needs further experimental and clinical research.

Supplemental Information

Supplemental Information 1 Supplementary materials and figures

Click here for additional data file.

Data S1 Raw Data

Click here for additional data file.

Supplemental Information 3 Original Western blot data

Click here for additional data file.

Supplemental Information 4 ARRIVE 2.0 Checklist

Click here for additional data file.

Supplemental Information 5 MIQE checklist

Click here for additional data file.

Thanks to Professor Sun Shuguo for his guidance and discussion of the manuscript.

Abbreviations

CHRAC1 chromatin accessibility complex subunit 1

YAP Yes-associated protein

IF Immunofluorescence

IHC Immunohistochemistry

CCK-8 Cell Counting Kit-8

GO Gene Ontology

KEGG Kyoto Encyclopedia of Genes and Genomes

GSEA Gene set enrichment analysis

OS overall survival

DFS disease-free survival

CESC Cervical squamous cell carcinoma and endocervical adenocarcinoma

BRCA Breast invasive carcinoma

KIRC Kidney renal clear cell carcinoma

ACC Adrenocortical carcinoma

Additional Information and Declarations

Competing Interests

Author Contributions

Human Ethics

Animal Ethics

DNA Deposition

Data Availability

The authors declare there are no competing interests.

Shasha Li conceived and designed the experiments, performed the experiments, analyzed the data, prepared figures and/or tables, authored or reviewed drafts of the article, and approved the final draft.

Lulu Wang performed the experiments, analyzed the data, prepared figures and/or tables, authored or reviewed drafts of the article, and approved the final draft.

Jing Shi analyzed the data, authored or reviewed drafts of the article, and approved the final draft.

Yi Chen performed the experiments, analyzed the data, prepared figures and/or tables, and approved the final draft.

Ang Xiao performed the experiments, prepared figures and/or tables, and approved the final draft.

Bingyue Huo performed the experiments, prepared figures and/or tables, and approved the final draft.

Wenjing Tian performed the experiments, prepared figures and/or tables, and approved the final draft.

Shilu Zhang performed the experiments, prepared figures and/or tables, and approved the final draft.

Gang Yang analyzed the data, authored or reviewed drafts of the article, and approved the final draft.

Wensheng Gong analyzed the data, authored or reviewed drafts of the article, and approved the final draft.

Huixia Zhang conceived and designed the experiments, performed the experiments, analyzed the data, prepared figures and/or tables, authored or reviewed drafts of the article, and approved the final draft.

The following information was supplied relating to ethical approvals (i.e., approving body and any reference numbers):

Cancer specimens involved in this study were approved by the Ethics Committee of of Huazhong University of Science and Technology and Shanghai Zhuoli Biotechnology Co., LTD (approval number: ZL2019-9-LL028, ZL2019-11-LL029).

The following information was supplied relating to ethical approvals (i.e., approving body and any reference numbers):

The animal experiments have been approved by the Ethics Committee of Huazhong University of Science and Technology (Approval number: 2022 IACUC Number: 3148).

The following information was supplied regarding the deposition of DNA sequences:

The RNA-seq data are available at FigShare: Li, Shasha (2023). Hela shCHRAC1. figshare. Online resource. https://doi.org/10.6084/m9.figshare.23989101.v1 and at the National Genomics Data Center: HRA004198.

https://ngdc.cncb.ac.cn/gsa-human/browse/HRA004198

The following information was supplied regarding data availability:

The raw data and original immunoblots are available in the Supplemental Files.

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
