# Peer review of "Chromatin accessibility complex subunit 1 enhances tumor growth by regulating the oncogenic transcription of YAP in breast and cervical cancer"

_PeerJ, doi:10.7717/peerj.16752_

## Round 0.1 · original submission · Minor Revisions

Dear Dr. Zhang,

Thank you for your submission to PeerJ.

It is my opinion, as the Academic Editor for your article - Chromatin accessibility complex subunit 1 enhances tumor growth by regulating the oncogenic transcription of YAP in breast and cervical cancer - that it still requires small Minor Revisions.

With kind regards,
Abhishek Tyagi
Academic Editor
PeerJ Life & Environment

Reviewer 1 ·

Basic reporting

The authors have a good writing style. The author followed an excellent article structure with a detailed data collection and methodology explanation. They attached their raw data to the supplementary files. I suggest that they should improve or rewrite the captions, especially with Figures 4, 5, and 6.

Experimental design

This research falls within the scope of PeerJ. They have a detailed description of the methodology. They use proper methods to investigate the protein interactions and protein expressions. The combination of Bio-ID, knockdown experiment, and NGS data are powerful and successfully revealed the function of CHRAC1.

Validity of the findings

The authors provide high-quality figures with good statistical analysis of their data. They spent enough time discussing the link between CHRAC1 and YAP in addition to the importance of their research. Overall, their conclusion should be reliable according to the provided data.

Additional comments

Overall, I suggest accepting this paper. In general, this is a high-quality research paper.
Several aspects of this work could be improved:
1. I suggest they improve the last paragraph of the introduction. They successfully pointed out the importance of the YAP pathway in the previous paragraphs but did not describe the importance of CHRAC1. They also need to spend more time discussing the connection between YAP and CHRAC1 and why they hypothesize there are interactions between YAP and CHRAC1.
2. I suggest they add more detail about the Bio-ID method. It’s good that they refer to their previous research but may need to describe more about this method in this paper as well.
3. For the statistical analysis methodology, I suggest they link their methodology with their figures. For example, they should mention that they used a T-test with colony formation in Figure 3B, etc.
4. There is a gap between the last part of the result and the discussion part. It feels like they need to add a summary paragraph to improve the transition.

·

Basic reporting

Good with no additional comments.

Experimental design

Good with no additional comments.

Validity of the findings

1. Figure 3C: please add a scale bar.

2. Figure 3E: please add # of n on the figure or in the legend

3. Figure 4D: is the p-value adjusted p-value? Please specify. Is the X-axis GeneRatio? Please add label.

4. Figure 4F: it would be better if you can show individual replicate value as dots on the bar plot. It would also be nice to do IHC & IF for some of the targets in cell line or human tissue to validate your finding at the protein level, one step further than the RNA level.

Additional comments

This article is clearly-written and the experiments are well-designed and executed. I'm in favor of accepting it after minor revisions stated in my comments.

---

## Round 0.2 · Major Revisions

Dear Dr. Zhang,

The Section Editors are concerned that the sole basis for demonstrating an interaction is the BioID data, and that this lacks controls. The authors should have attempted to validate this by some other means and this should have been picked up earlier in the review process.

The authors say: “Methods. Bio-ID method was used to identify the interactome of transcriptional activator Yes-associated protein (YAP) and the binding between YAP and CHRAC1 was verified by immunofluorescence. MTT, colony formation and subcutaneous xenograft assays were conducted to explore the function of CHRAC1 in cancer cell proliferation. RNA-seq analysis and RT-PCR were used to analyze the transcription program change after CHRAC1 ablation. The diagnostic value of CHRAC1 was analyzed by TCGA database and further validated by immunohistochemistry staining.”

For this type of study, merging IF images of 2 proteins is not sufficient; the authors should have attempted another verification method, such as cross-linking, duolink, co-immunoprecipitation, or similar.

Further, it is not clear that the authors have provided the data for all of the approaches and assays employed.

The controls are very weak in many cases, for instance, the paratumor does not look like normal tissue. This histology staining is of poor quality and the authors need to elaborate on how they got to those graphs.

siRNA downregulation should be confirmed by immunoblot, and results compared to a distinct family member. The study is crying out for an over-expression comparison in the same cells.

RNAseq details are missing: how many replicates, reading to what depth, etc. Just reporting the RNAseq power is not a sufficient level of detail.

Please address all of these in a new revision.

With kind regards,
Abhishek Tyagi
Academic Editor
PeerJ Life & Environment

·

Basic reporting

Good with no additional comments.

Experimental design

Good with no additional comments.

Validity of the findings

The authors addressed my previous comments well and I have no additional comments.

---

## Round 0.3 · Minor Revisions

Dear Dr. Zhang,

Thank you again for submitting your revised manuscript. We sincerely commend the authors for their diligent efforts in refining the manuscript based on the editor's comments. However, upon reviewing the revised version, we observed that CHRAC1, being a nuclear protein, is expected to exhibit selective expression in the nucleus as showed in cultured cell lines data in Figure 5B. Unfortunately, the provided immunohistochemistry (IHC) data for breast and cervical para-tumor and tumor tissues does not align with this expectation and mostly showed diffused cytoplasmic positivity both in the tumor as well as in non-tumor cells. In light of this, we kindly request the authors to furnish higher magnification and contrasting images for all IHC data to thoroughly examine and address this to rule out any potential discrepancies.

Additionally, we advise the author to label the stage of the tumor tissue of each primary tumor tissue for Figure 2A and Figure 5F with higher magnification and contrasting images.

Without substantial revisions, we will be unlikely to make a decision and would encourage you to address them in full. Please highlight all changes in the manuscript text file.


With kind regards,
Abhishek Tyagi
Academic Editor
PeerJ Life & Environment

---

## Round 0.4 · accepted · Accept

Dear Dr. Zhang,
Thank you for your revised submission. It was satisfactory.
I am writing to inform you that your manuscript - Chromatin accessibility complex subunit 1 enhances tumor growth by regulating the oncogenic transcription of YAP in breast and cervical cancer - has been Accepted for publication.

Congratulations, and thank you for your submission to PeerJ.

With kind regards,
Abhishek Tyagi
Academic Editor
PeerJ Life & Environment